# Both conventionally and organically fertilized tomatoes maintain fruit quality through uncontrolled green peach aphid infestation, with a transcriptional shift towards catabolism

June Labbancz[1], Luke Gustafson[2¤], Preston Andrews[2], Amit Dhingra[1*]

**1** Department of Horticultural Sciences, Texas A&M University, College Station, Texas, United States of America, **2** Department of Horticulture, Washington State University, Pullman, Washington, United States of America

¤ Current Address: The Davey Tree Company, Baltimore, Maryland, United States
* adhingra@tamu.edu

## Abstract

Aphids are a major pest of greenhouse-grown temperate crops, responsible for billions in crop damage yearly. As organic agriculture rapidly grows in popularity, understanding how plants grown under organic systems respond to insect pest pressure may give insights into better management practices and information about the genes of interest for crop improvement. We measured the response of tomato (*Solanum lycopersicum*) leaf and fruit transcriptome, as well as a few selected metabolites in the mature fruit, to an infestation of the generalist green peach aphid (*Myzus persicae*). The aphids were introduced approximately halfway through the lifecycle of the plants that were grown under conventional and organic fertilizer regimes. While plants provided with conventional fertilizer experienced greater aphid infestation, neither group suffered a significant loss in total yield or fruit quality. This result is likely a consequence of ample nutrient and water availability. Co-expression network analysis using WGCNA revealed that in leaf tissue, both treatment groups showed a general shift from diverse anabolic processes to catabolism, while fruit tissue experienced relatively minor changes. At the stage of infestation investigated, abscisic acid appeared to be the main phytohormone response. One coexpression network module showed a correlation with both organic fertilizer treatment and aphid infestation; its hub gene (Solyc02g078940.3) may be of interest in exploring unique responses to phloem feeding insect infestation under an organic fertilizer regime.

## Introduction

Organically grown crops are experiencing rapid growth in popularity, with a global retail value exceeding $119 billion [1]. However, crops grown in organic systems have lower total yields than their conventionally grown counterparts [2]. Generally,

---

**Data availability statement:** All RNA sequencing files are available at the National Center for Biotechnology Information Short Read Archive (NCBI SRA) repository under BioProject PRJNA1329212 (https://www.ncbi.nlm.nih.gov/bioproject/PRJNA1329212).

**Funding:** This research was funded in part by CSANR BIOAg grant to PA and AD. USDA National Institute of Food and Agriculture, Hatch projects WNP00011 and Texas A&M AgriLife Hatch Project #TEX0-9950-0 and start-up funds from Texas A&M AgriLife Research and Texas A&M University to AD. JL acknowledges graduate research assistantship support from the Department of Horticultural Sciences at Texas A&M University.

**Competing interests:** The authors have declared that no competing interests exist.

agricultural products grown under organic management systems have compositional differences from conventionally grown plants, including increased antioxidants, phenolics, carotenoids, vitamin C, and decreased pesticide residues [3–7]. The majority of meta-studies have found evidence to support the claim that organic crops contain higher quantities of select nutrients compared to conventionally grown crops [8]. There is also a benefit to producers in cultivating organic crops, as with only modest price premiums, organically grown crops generate more value for growers than conventional crops [9].

Given the growing economic significance of organic agriculture, it is worthwhile investigating how organic crops respond to environmental stresses. Aphids are among the most destructive agricultural pests in temperate climates, causing losses of millions of metric tons of crops annually [10]. Using sucking mouthparts called stylets to siphon off phloem sap, they can deprive plants of nutrients, manipulating plant signaling pathways, and introducing viral disease to host plants [10–12]. The limiting factor in aphid growth is the availability of essential amino acids in the relatively nitrogen-poor phloem sap [13]. Aphids excrete an incompletely digested sap known as honeydew, which can further encourage mold growth [10,14]. When combined with the rapid parthenogenic reproduction that aphids exhibit during summer months, populations can quickly strain crops, and containment of aphid infestations typically relies on the application of pesticides [10]. Organic agriculture has a limited number of approved pesticides, and especially in environments such as greenhouses where natural predators may be excluded and warmth is maintained year-round, a multitude of control measures, including netting, parasitoid and predator releases, and biopesticide applications, may be necessary [15].

Despite the economic impact of aphids, their interactions with host plants remain incompletely understood due to the complexity of these interactions. Resistance genes that can limit aphid infestation in tomato plants have been reported [16,17], but this is only helpful for a limited subset of cultivars. Many phytohormone responses have been associated with aphid feeding, with salicylic acid appearing to be the most common response, but jasmonic acid and ethylene appear to elicit a more successful defense against aphid feeding [18–23]. Evidence regarding the role of abscisic acid appears to be more inconsistent [24,25]. Generation of reactive oxygen species (ROS) can successfully limit aphid feeding [26,27] but generally requires the plant to protect itself by generating antioxidant compounds. In tomato, one of the most abundant antioxidants is the carotenoid lycopene [28]. Still, a wide range of antioxidants, including catalase, peroxidase, and myriad flavonoids and phenolic compounds, help in allowing the plant to survive its ROS-based resistance to aphid feeding. Other secondary metabolites, which are elicited by aphid feeding, may aid in resistance, including cardiac glycosides, alkaloids, benzoxazinoids, and glucosinolates [29]. Tomato plants may also use volatile compounds, often terpenoids, to attract aphid predators and parasitoids [30,31].

Previously, we reported that tomatoes grown in organic growing systems display major changes in gene expression, which manifest as differences in secondary metabolite content, including phenolic compounds, ascorbate, and lycopene [7].

Several of these desirable compounds have potential benefits in human nutrition [32]. Alteration of fertilizer input can create major global changes in gene expression.

Nitrogen is central to plant metabolism, and its form and availability impact most aspects of physiology [7,33–35]. Organic fertilizers generally provide nitrogen in the form of ammonia, and release is slower, depending on environmental conditions [36]. Physiological differences resulting from the nitrogen source can alter how plants respond to aphid feeding in organic systems compared to conventional ones. Soil microbial communities may be altered, generating further physiological responses [37]. Organic fertilizer application has been shown to alter pest success [38,39], though the precise molecular mechanisms of these differences remain unclear.

To explore the interaction between extensive aphid infestation and organic production, we designed a study in which tomato plants were provided organic fertilizer or conventional fertilizer and were subjected to aphid infestation or kept aphid-free. We hypothesized that aphid infestation would result in significant alterations to final fruit quality, and that specific genes would show a distinct correlation with both fertilizer type and the presence of aphids, indicating growing environment-specific responses to aphid feeding. At harvest, biochemical analysis of sampled tissues demonstrated the impacts of aphids on plant health and fruit quality. Quantitative RNA sequencing of fruit and leaf tissues enabled the Weighted Gene Co-Expression Network Analysis (WGCNA) to identify clusters of genes associated with the two experimental treatments.

## Materials and methods

### Growth conditions and sample collection

As reported previously [7], the plants used for this study were Oregon Spring tomatoes (*Solanum lycopersicum* cv. 'Oregon Spring'), a determinate, early-season tomato cultivar. Plants were randomly assigned to 4 groups by two variables (aphid infestation vs control and conventional vs organic) and grown in #7 pots in a greenhouse under insect netting with supplemental 1000W metal halide lighting. The organic group was planted in 75:20:5 LC1 potting soil: Whitney Farms Organic Potting Soil: Topsoil, while the conventional group was planted in 100% LC1 potting soil. Biolink 5-5-5 fertilizer and micronutrient mixture was applied to the organic group, and Peters 20-10-20 and superphosphate was applied to the conventional group at a consistent rate of total nitrogen. Pathogen-free green peach aphids (*Myzus persicae*) were reared by the Washington State University Department of Entomology on *Brassica oleracea* prior to transfer to tomato plants. At 14 weeks of growth, 100 wingless aphids were added to aphid+ group plants. At week 16, an additional 300 aphids were added to each plant in the aphid+ group. The harvest of fruit at the red ripe stage began at week 18, until at least eight fruits were collected from each plant at week 23. At the time of harvest, the number of aphids present on 3–4 representative leaf subsamples per plant were counted using a hand lens for magnification, and the number of aphids per leaf mass was calculated. For fruit samples, a one-centimeter-thick slice was excised from the equatorial region of each fruit, and the pericarp tissue external to each locule was sampled. The tissue to be used for metabolic analysis and RNA isolation was flash-frozen with liquid nitrogen, placed in 50-mL plastic tubes, and stored in a −80 °C freezer. At week 23, three to four randomly selected representative leaves were also sampled from the mid to upper canopy of all plants and flash-frozen with liquid nitrogen. The frozen samples were pulverized using a SPEX SamplePrep Freezer Mill 6870 before RNA isolation and RNAseq.

### Biochemical analysis

Nitrogen, Carbon, Calcium, Potassium, Magnesium, Sodium, Phosphorus, and Sulfur content were determined by ICP-MS (Analytical Science Laboratory, University of Idaho, Moscow, ID), preceded by combustion (for carbon and nitrogen) or nitric acid digestion (all others). Total phenolic content was assessed by spectrophotometry as outlined previously [40] with modifications. 350 mg of powdered, frozen fruit or 150 mg of powdered, frozen leaf was added to 1 mL of 80%

methanol solution, in triplicate for each sample. Tubes were vortexed for 30 seconds before 24-hour storage at –20 °C. Samples were then centrifuged at 8,000g for 20 minutes at 4 °C. The supernatant was transferred to a fresh tube and adjusted to 3 mL with 80% methanol. 200 μL of extract was added to 1mL of 10% Folin-Ciocalteu (F-C) phenol reagent and 800 μL g/L sodium carbonate, then vortexed and stored at 20°C for 2 hours. An HP8453 ultraviolet-visible spectrophotometer (Agilent Technologies, Santa Clara, USA) was blanked on 1 mL 10% F-C reagent in a 1.5 mL polystyrene cuvette. 1 mL of sample was measured in duplicate for each sample at 760nm. Absorbance measurements were compared to a standard curve for gallic acid and reported in gallic acid equivalents. Lycopene, beta-Carotene, and chlorophyll content were assessed via spectrophotometry as outlined by [41] with modification. 8.33 mL of 2:3 acetone: hexane (HPLC grade), amended with 200 mg/L butylated hydroxytoluene at 4 °C, was added to 100 mg frozen milled fruit, then incubated at −20 °C for 96 hours with 1-minute vortexing every 24 hours. The spectrophotometer was blanked using HPLC-grade hexane. The hexane phase of each sample was transferred to a 1.4 mL glass cuvette, and 505nm absorbance was measured. Trolox equivalent antioxidant capacity (TEAC) was determined as in [42] with modifications. 700 μL of 50 mM MES buffer and 700 μL of 100% ethyl acetate were added to 200 mg frozen powdered tissue. The samples were vortexed for 30 seconds, then centrifuged at 8000g for 10 minutes at 4 °C. The organic and aqueous phases were pipetted into separate tubes and centrifuged again at 8000g for 10 minutes at 4 °C. The spectrophotometer was blanked with 1 mL 100 mM phosphate buffer (pH 7.4) in a 1.4 mL glass cuvette at 734 nm. For hydrophilic TEAC (HTEAC), 40 μL Hydrogen Peroxide, 100 μL 15 mM 2,2'-azino-bis-(3-ethylbenzthiazoline-6-sulfonic acid) (ABTS), and 10 μL of 3.3 units/μL peroxidase from horseradish (HRP) were added to cuvettes and shaken for 10 seconds. 800 μL of 100 mM phosphate buffer (pH 7.4) was added to the cuvettes and gently shaken for 10 seconds. Cuvettes were loaded into the spectrophotometer multi-cell transport, and a timed kinetic assay was run at 734nm. After two passes, 50 μL MES extract was added and mixed, and the decrease in absorbance was measured. For lipophilic TEAC (HTEAC), the same protocol was repeated, replacing 800 μL 100mM phosphate buffer with 770 μL 100% ethanol and 50 μL MES extract with 80 μL ethyl acetate extract. Standard curves for LTEAC and HTEAC were produced by replicating the protocol above, substituting powdered tissue with solutions of Trolox in MES buffer or ethyl acetate. Ascorbic acid content was measured as in [43]. 1.5 mL of 1 M HClO4 at 4 °C was added to 200 mg of frozen, powdered tissue. Samples were vortexed for 30 seconds, incubated on ice for 20 minutes, then centrifuged at 8000g for 10 minutes at 4 °C. 400 μL of supernatant was transferred to a new tube, and 200 μL of 100 mM 2-ethanesulfonic acid (HEPES-KOH buffered, pH 7.0) was added, followed by 30 seconds of vortexing. Samples were titrated to pH 4–5 with 5M potassium carbonate, followed by centrifugation at 8000g for 5 minutes at 4 °C. The spectrophotometer was blanked at 265nm with 500 μL 100mM phosphate buffer (pH 5.6) in a 0.5 mL quartz cuvette. 50 μL of extract was added to 446 μL 100mM phosphate buffer (pH 5.6) in a 0.5mL quartz cuvette and mixed. Cuvettes were loaded into a multi-cell transport, and a timed kinetic assay was run. After two measurements, 4 μL of 1 unit/μL ascorbate oxidase from *Cucurbita* sp. was stirred into each cuvette, then the decrease in absorbance was measured. For total ascorbic acid measurements, the same protocol was followed with modifications. After titration, 200 μL of extract was added to 400 μL 100mM phosphate buffer (pH 5.6) with the addition of 31.8 μL 1M dithiothreitol. Tubes were vortexed for 10 seconds and stored on ice for 5 minutes. 100 μL of solution and 396 μL 100mM phosphate buffer (pH 5.6) were mixed, and measurement was continued as with the reduced ascorbic acid assay. Standard curves were produced using the above protocol, substituting tissue for solutions of dehydro-L-(+)-ascorbic acid dimer.

## RNA sequencing

Samples were collected from three biological replicates for each of the eight groups (2 aphid statuses x 2 fertilizer treatments x 2 tissue types); RNA was isolated from pulverized leaf and fruit tissues at the red ripe stage using a QIAGEN RNeasy Plant Mini kit. Samples were treated with DNase to remove DNA contamination. RNA concentration was assessed with a Nanodrop ND-8000, and integrity was assessed via agarose gel electrophoresis. Extracted RNA was used to generate barcoded samples for Illumina sequencing using the TruSeq RNA Sample Preparation v2 kit. The

quantity and quality of all samples were evaluated using Life Technologies Qubit Fluorometer and an Agilent 2100 Bioanalyzer. All cDNA libraries were sequenced as 2x100PE reads across three lane flow cells using an Illumina HiSeq 2000 sequencing platform.

### Data analysis

Significance of differences of biochemical and agronomic traits between groups were assessed by two-way ANOVA using Statistical Analysis system v.9.1.3 (SAS Institute, Incorporated, Cary, North Carolina). Natural log transformation was used when data did not meet normality or equal variance assumptions. Aphid counts between the organic and conventional groups were assessed for significance by a two-sample t-test. A $p < 0.05$ threshold was selected for determining significance in all tests.

Sequencing reads were assessed for quality using FastQC (version 0.11.9). The reference genome used was *Solanum lycopersicum* SL3.0 from Ensembl. Reads were aligned to the reference genome using CLC 23.0.5 with default parameters, and TPM-adjusted expression levels were recorded for all genes. WGCNA analysis was performed in R on log2-normalized data [44]. Leaf and fruit tissue samples were used to create two separate co-expression networks. Soft thresholding powers of 4 and 10 were selected for leaf and fruit tissue, respectively, based on the lowest value that surpassed a scale-free topology model fit of 0.8. A signed network with a maximum block size of 10000 and a minimum module size of 30 was produced for each dataset, otherwise using default parameters. Reference gene names were accessed from DAVID (version 2021). Functional annotation using DEEP-GO SE [45] with default parameters was used to create an annotation list for functional enrichment analysis using topGO with the test parameter "Fisher classic".

## Results and discussion

### Aphid and phytonutrient analysis

Aphid counts were approximately 4 times higher on aphid-infested plants under conventional fertilizer treatment than on aphid-infested plants grown under organic fertilizer treatment. Between aphid introduction at flowering and harvest, aphid populations increased ~4x on ORG+ plants and ~20x on CONV+ plants (Table 1). Despite the differences in aphid infestation, yield, which was higher in plants treated with conventional fertilizer, was not significantly impacted by the presence of aphids (Table 1). Similarly, plant carbon, nitrogen, and mineral content were altered by fertilizer type, but not by the presence of aphids. Plants treated with organic fertilizer had reduced carbon, nitrogen, and phosphorus content, and elevated mineral content, relative to conventionally fertilized plants (Table 1). While vegetative tissue and agronomic performance were not significantly affected by the aphid infestation, some significant responses were seen in measures of fruit quality (Table 2). Ripe fruit phenolic content was elevated by aphid infestation and by organic fertilizer. In contrast, a fertilizer type by aphid interaction was seen in antioxidant activity, with the organic fertilizer treatment group seeing a modest increase, and the conventional fertilizer group seeing a modest decrease. Brix, however, was only responsive to the type of fertilizer, not to aphids, with organic fertilizer leading to an increase. Sugar, lycopene, and phenolic compounds are all desirable attributes in tomato fruit; organic fertilizer improved fruit quality by every metric tested, and contrary to expectation, aphid infestation seems to have enhanced fruit phenolic content while otherwise having no apparent detrimental effect (Table 2).

### WGCNA and biochemical pathway analysis

Co-expression networks were created for both leaf and fruit tissue, with a total of 54 modules generated for leaf samples and 57 modules generated for fruit samples (Figs 1 and 2, respectively). Modules with significant correlations with both aphid infestation and fertilizer type were selected for further analysis from the leaf and fruit-derived networks (Figs 3 and 4, respectively). In leaf tissue, modules yellow, green, brown, greenyellow, and red fit these criteria, while in the fruit tissue grey60 and navajowhite2 fit these criteria. Additionally, modules blue and turquoise in the leaf co-expression network were

**Table 1. Assessed chemical characteristics of tomato plants; *=p<.05, **=p<.01, ***=p<.001, ns=not significant, NA=not applicable. A total of 24 plants in 4 groups were assessed. Significance for aphid counts was determined by two sample t-test, while significance for all other traits was determined by two-way ANOVA.**

| | Units | CON- | CON+ | ORG- | ORG+ | Fertilizer | Aphid | Fertilizer*Aphid |
|---|---|---|---|---|---|---|---|---|
| Aphid Count | aphids/g FW | | 4.68 | | 1.06 | * | NA | NA |
| Yield, ripe fruit | kg | 1.62 | 1.77 | 1.64 | 1.37 | ns | ns | ns |
| Yield, green fruit | kg | 0.9 | 0.67 | 0.55 | 0.58 | * | ns | ns |
| Yield, all fruit | kg | 2.52 | 2.44 | 2.19 | 1.95 | *** | ns | ns |
| Vine FW | kg | 1.33 | 1.3 | 1.15 | 1.22 | ns | ns | ns |
| Root FW | g | 36.99 | 35.76 | 32.31 | 37.88 | ns | ns | ns |
| Leaf Phenolics Conc. (leaf) | mg GA equiv. / g FW | 0.83 | 0.72 | 0.87 | 0.82 | ns | ns | ns |
| Leaf Carbon | % DW | 38.33 | 38.33 | 35 | 34.67 | *** | ns | ns |
| Leaf Nitrogen | % DW | 5.1 | 5.23 | 4.1 | 4.1 | *** | ns | ns |
| Leaf Calcium | ug / g DW | 42000 | 37000 | 47000 | 47000 | *** | ns | ns |
| Leaf Potassium | ug / g DW | 56333.33 | 60666.66 | 64000 | 65000 | ns | ns | ns |
| Leaf Magnesium | ug / g DW | 9033.33 | 7900 | 9733.33 | 9766.66 | * | ns | ns |
| Leaf Sodium | ug / g DW | 910 | 1040 | 993.33 | 866.66 | ns | ns | ns |
| Leaf Phosphorous | ug / g DW | 14333.33 | 14000 | 12000 | 11666.67 | * | ns | ns |
| Leaf Sulfur | ug / g DW | 14000 | 18333 | 36333.33 | 35666.66 | *** | ns | ns |

**Table 2. Assessed chemical characteristics of red ripe fruit; *=p<.05, **=p<.01, ***=p<.001, ns=not significant. For aphid count, yield, vine and root fresh weight, and leaf phenolics content, 24 plants in 4 groups were assessed. For leaf carbon, nitrogen, calcium, potassium, magnesium, sodium, phosphorus, and sulfur, 12 plants in 4 groups were assessed. Significance for all traits were determined by two-way ANOVA. Where log transformation was necessary, "Ln" is noted in the units column.**

| | Units | CON- | CON+ | ORG- | ORG+ | Fertilizer | Aphid | Fertilizer*Aphid |
|---|---|---|---|---|---|---|---|---|
| Brix | Ln °Brix | 1.6299 | 1.607 | 1.7644 | 1.8278 | *** | ns | ns |
| Phenolics Conc. Corrected for AsA (fruit) | mg GA equiv. / g FW | 20.9954 | 23.0282 | 24.1566 | 27.2262 | *** | * | ns |
| Hydrophilic TEAC | Ln mmol / g FW | 4.7976 | 4.7603 | 4.8619 | 4.955 | *** | ns | * |
| Lypophilic TEAC | mmol / g FW | 1.4576 | 1.504 | 1.6567 | 1.5775 | *** | ns | ns |
| Lycopene | ug / g FW | 3.7107 | 3.8447 | 4.0579 | 4.3826 | *** | ns | ns |
| Reduced ascorbic acid | ug/g FW | 159.49 | 154.47 | 173.17 | 177.51 | *** | ns | ns |

selected for analysis based on their size and extremely strong correlation with aphid infestation, indicating the shared responses of the tomato plants to aphids, regardless of the fertilizer regime. Genes with the highest degree of module membership were selected as hub genes, candidate drivers of module function (Tables 3 and 4). Gene Ontology (GO) enrichment analysis was performed on these modules to identify commonalities in gene function within them, which may be essential for understanding differential responses to aphid infestation in organic agriculture.

Leaf tissue, the site of aphid feeding, exhibited a far greater response to aphid infestation. The two largest modules of the co-expression network, turquoise (5550 genes) and blue (5020 genes), are almost perfectly correlated with an aphid-free and aphid-infested state, respectively. Module turquoise is enriched in a wide variety of developmental and anabolic processes, including cell cycle-related functions and auxin response (Fig 5). Module blue is enriched in catabolic processes, reflecting the plant tapping into stored resources to maintain physiological functions (Fig 6). The only enriched hormone-related function in module blue is abscisic acid response, which is ineffective in reducing aphid feeding, but may aid in surviving the abiotic stresses that result from heavy aphid feeding.

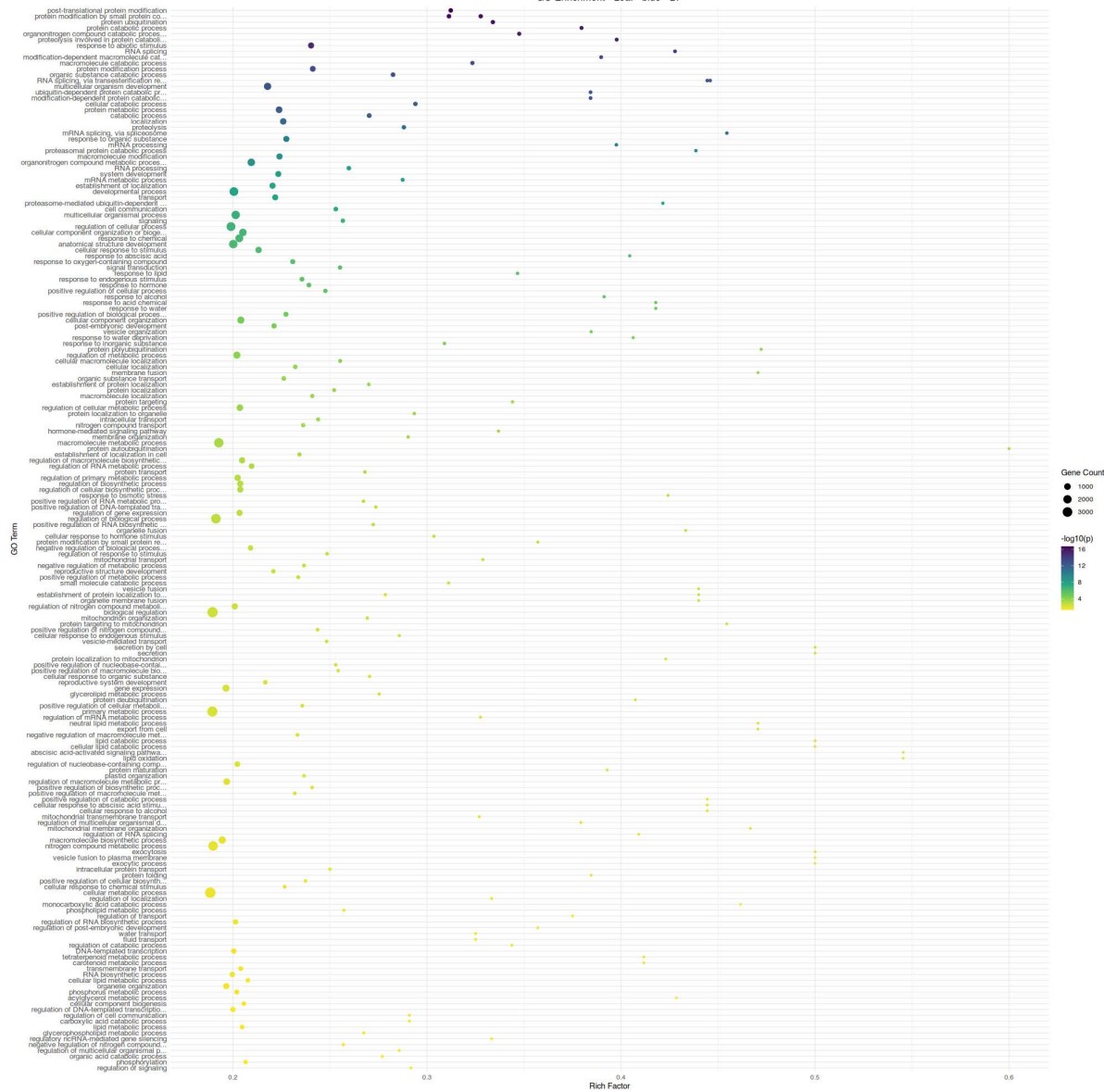

**Fig 1. Cluster dendrogram describing the gene co-expression network generated from leaf samples in this study.** Modules were identified using the dynamic tree cutting algorithm with a minimum module size of 30 genes and a merge cut height of 0.25.

Module yellow, the module associated with both organic fertilizer treatment and aphid infestation, appears to be enriched mainly in nitrogen-associated metabolic processes, as well as other central metabolic pathways (Fig 7). The hub gene of this module (Solyc02g078940.3) is a PIG3 homolog quinone oxidoreductase (Table 3); homologs in maize have demonstrated function in resistant insects and fungi through the generation of reactive oxygen species [46]. As the plants grown under organic fertilizer were able to maintain an aphid population approximately four times lower than those under conventional fertilizer, this gene and other genes within this module may be of interest in examining resilience to sustained

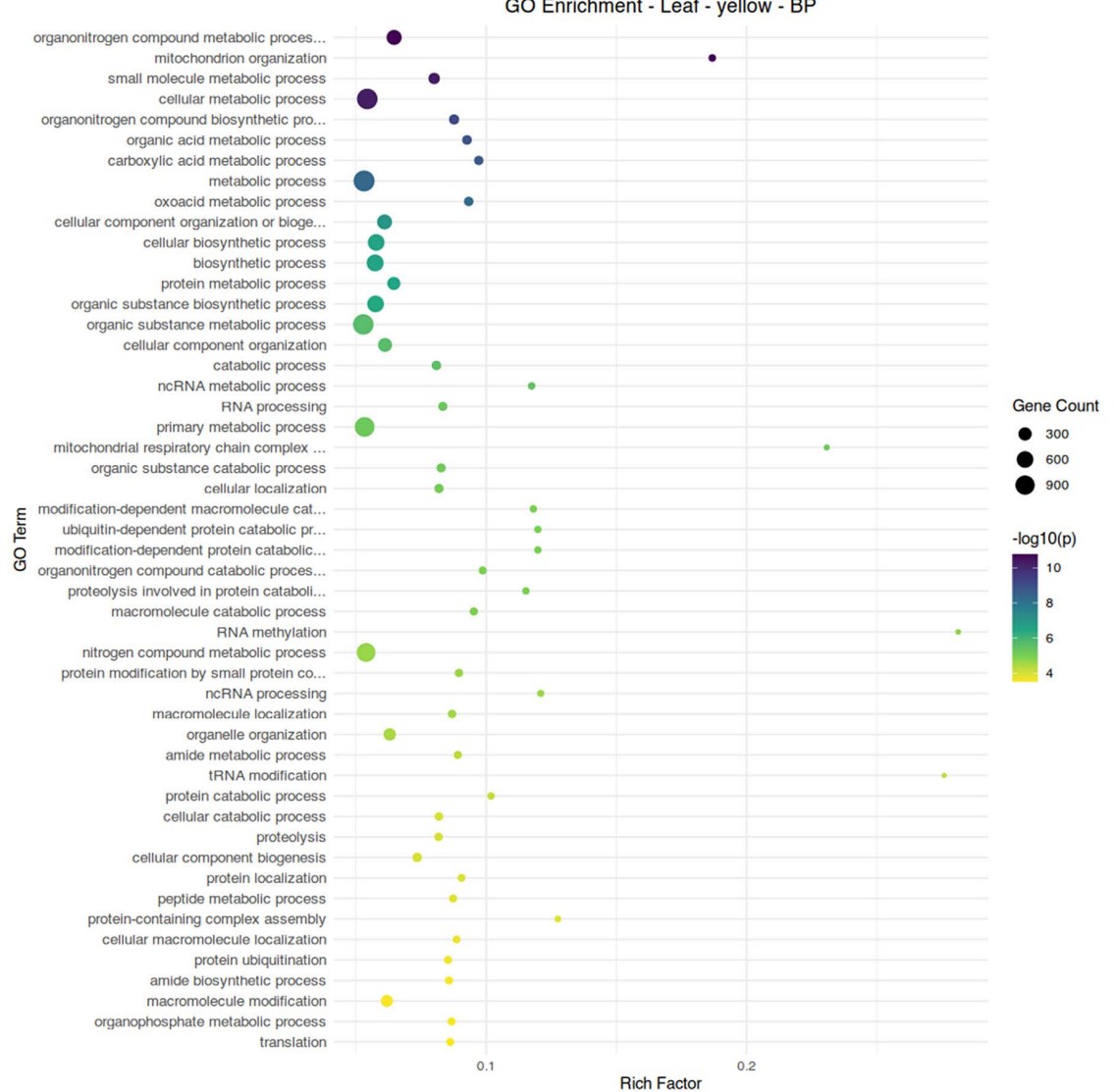

**Fig 2. Cluster dendrogram describing the gene co-expression network generated from fruit samples in this study.** Modules were identified using the dynamic tree cutting algorithm with a minimum module size of 30 genes and a merge cut height of 0.25.

aphid feeding. Module green, the module associated with conventional fertilizer and aphid infestation in leaf tissue, has 1,3-β-Glucan Synthase (Solyc07g056260.3) as its hub gene. The deposition of callose on sieve plates is a known response to feeding, effectively sectioning off portions of the plant vascular system to limit damage. While callose is a mechanism for responding to stress, it can exert regulatory effects itself [47]. As aphid populations were high on plants treated with conventional fertilizer, this response may reflect a suppression of plant defenses outside of direct responses to wounding.

Two modules, red and greenyellow, showed a negative association with aphid infestation and an association with conventional fertilizer treatment. While module red shows diverse functions related to cellular development and biosynthesis, module greenyellow clearly shows functionality centered around the chloroplast and photosynthetic processes (Figs 8-10).

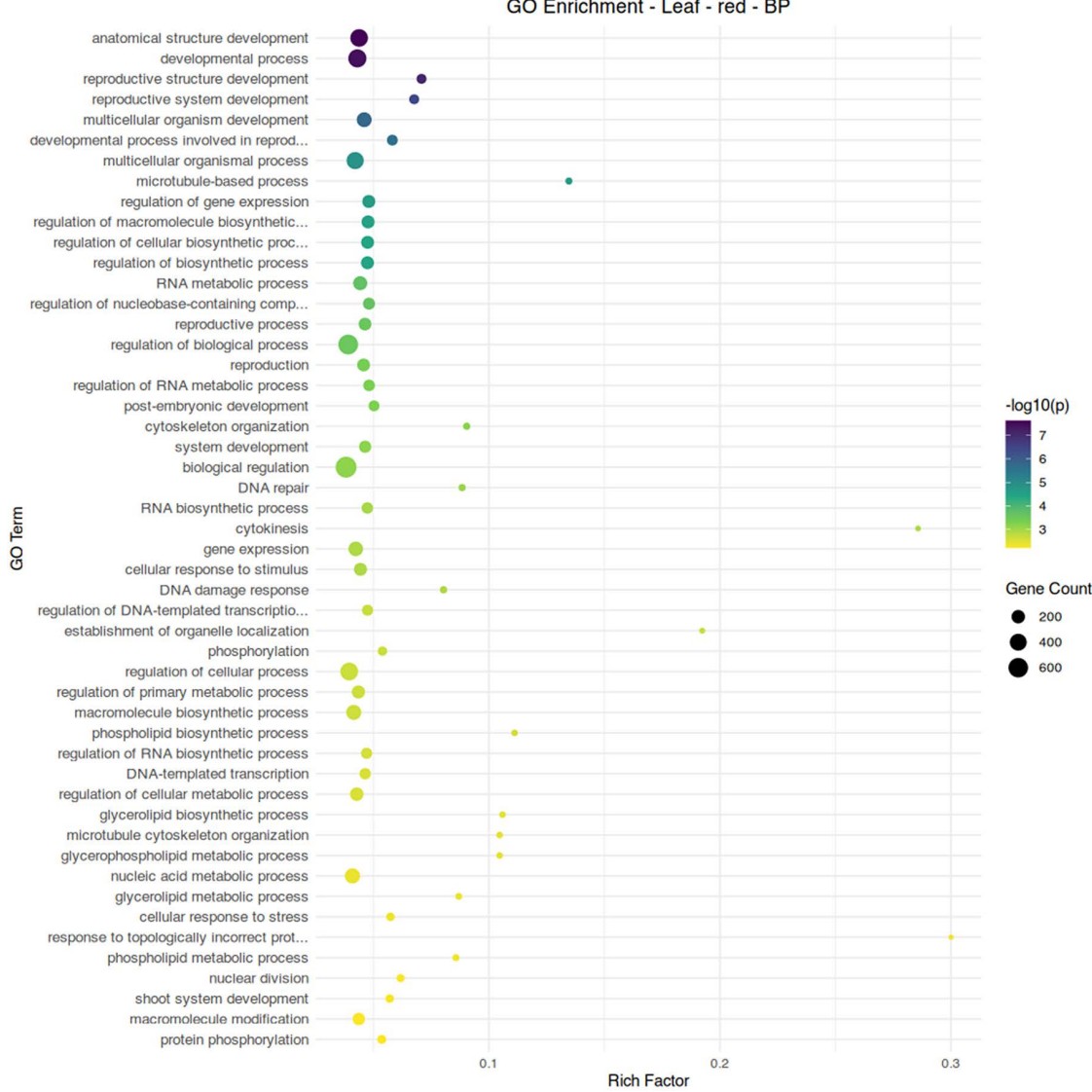

**Fig 3. Correlations between modules and aphid presence and organic fertilizer treatment in the gene co-expression network generated from leaf samples.** *=p<0.5, **=p<0.1, ***=p<0.01.

Module brown, associated with aphid-free plants under organic fertilizer, also shows an abundance of chloroplast-specific genes, but with functionality more strongly associated with biosynthetic reactions (Fig 10). Overall, suppression of chloroplast functions appears to be a common element of aphid infestation in both organic and conventional fertilizer regimes.

Simple resource deprivation may also limit aphid population growth. Plants treated with organic fertilizer had significantly lower leaf nitrogen content, about 80% of the levels of those in the conventional treatment group (Table 1). The difference in aphid populations between treatment groups may reflect plant nitrogen status as much as elicited defense responses. Aphids are dependent upon their host plant and are generally limited by plant nitrogen content, showing a strong response to host nitrogen content and host nitrogen fertilization [13,48,49]. Fertilizer type can influence aphid abundance as well, with differences between mineral fertilizer, manure, and legume crop rotation creating differences in both

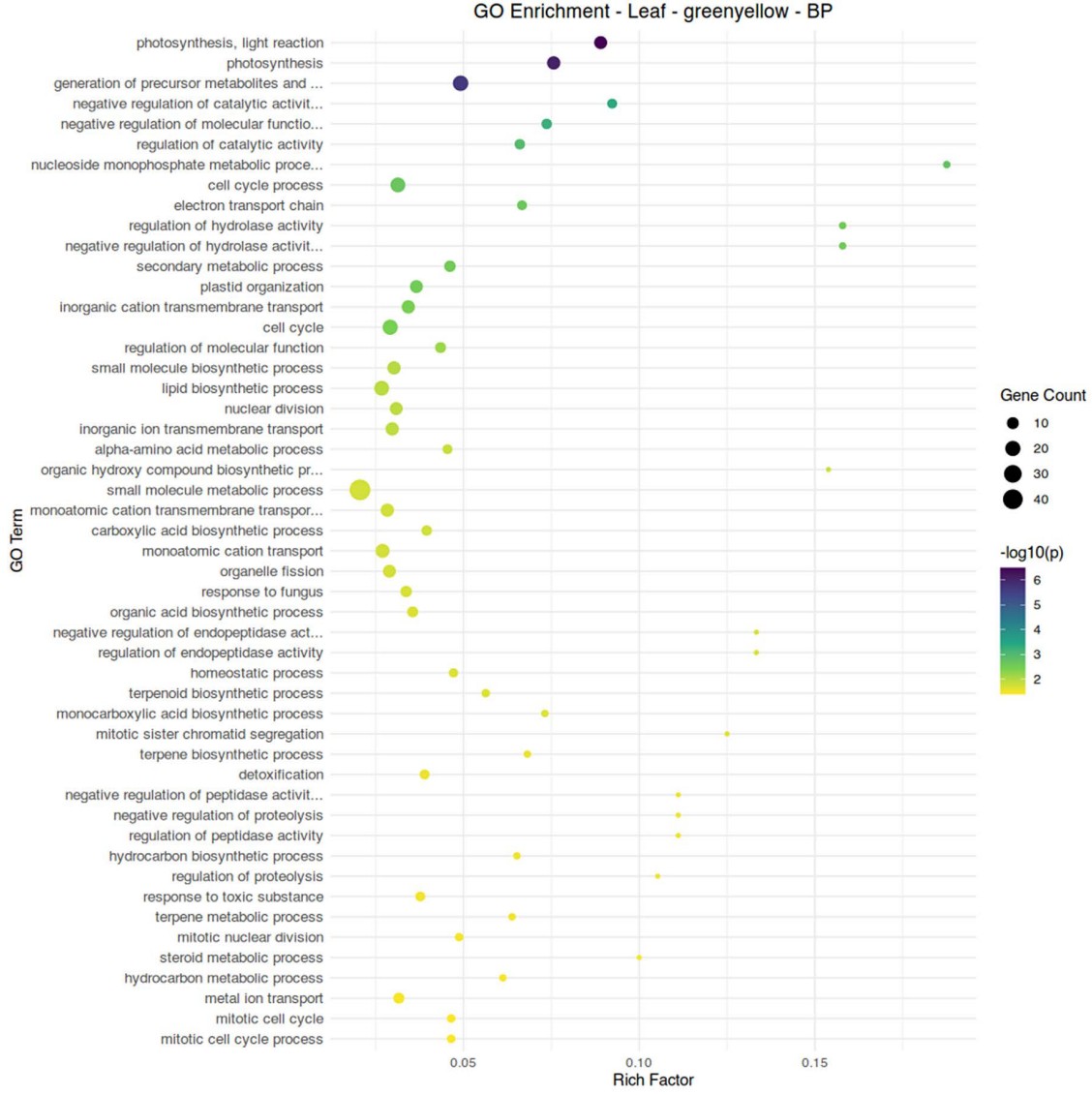

**Fig 4. Correlations between modules and aphid presence and organic fertilizer treatment gene co-expression network generated from fruit samples.** \*=p<0.5, \*\*=p<0.1, \*\*\*=p<0.01.

abundance and the species of aphids present under field conditions [50]. The relationship between nitrogen availability and aphid abundance is not linear. However, fields fertilized by legume crop rotation providing less nitrogen than conventional mineral fertilization enable greater quantities of the aphid *Sitobon avenae* on wheat (*Triticum aestivum*). The apparent negative relationship between organic fertilization and the abundance of the aphids in this study may differ depending upon the species of aphid and the specific organic fertilizer provided. Additionally, while decreased nitrogen fertilization may lead to lower pest populations, this does not necessarily amount to a net positive effect on plant health [51].

Fruit tissue manifested far milder transcriptional responses to aphid feeding, as this is not the site of aphid feeding. This was reflected in the limited impact of aphid feeding on fruit quality (Table 2). Modules with significant correlations were primarily responsive to fertilizer treatment, with only three relatively small modules showing a response to aphid

**Table 3. Select modules from leaf co-expression network. *=p<.05, **=p<.01, ***=p<.001, ns=not significant.**

| Gene | Gene Name | Module | Module Association | Module Membership | Aphid | Organic |
|---|---|---|---|---|---|---|
| Solyc01g103990.3 | uncharacterized LOC101250254 (LOC101250254) | blue | Aphid | 0.996 | 0.970*** | −0.024 |
| Solyc03g117360.3 | Uncharacterized protein (A0A3Q7GI15_SOLLC) | turquoise | No Aphid | 0.994 | −0.943*** | −0.029 |
| Solyc02g078940.3 | quinone oxidoreductase PIG3 (LOC101268548) | yellow | Aphid, Organic | 0.967 | 0.650*** | 0.407** |
| Solyc07g056260.3 | 1,3-beta-glucan synthase (A0A3Q7HGL2_SOLLC) | green | Aphid, Conventional | 0.938 | 0.852*** | −0.264 |
| Solyc03g097470.3 | 3-oxoacyl-[acyl-carrier-protein] synthase 3 A, chloroplastic (LOC101249057) | brown | No Aphid, Organic | 0.981 | −0.542*** | 0.545*** |
| Solyc02g068080.3 | chloride channel protein CLC-b (LOC101249604) | red | No Aphid, Conventional | 0.942 | −0.576*** | −0.405** |
| Solyc01g080280.3 | glutamine synthetase (GS2) | greenyel-low | No Aphid, Conventional | 0.936 | −0.604*** | −0.440*** |

**Table 4. Select modules from fruit co-expression network. *=p<.05, **=p<.01, ***=p<.001, ns=not significant.**

| Gene | Gene Name | Module | Module Association | Module Membership | Aphid | Organic |
|---|---|---|---|---|---|---|
| Solyc02g090890.3 | Zeaxanthin epoxidase, chloro-plastic (A0A3Q7FC15_SOLLC) | grey60 | Aphid, Conventional | 0.964 | 0.465*** | −0.623*** |
| Solyc02g085870.3 | 3-ketoacyl-CoA synthase 6(CER6) | nava-jowhite2 | No Aphid, Conventional | 0.916 | −0.301 | −0.357* |

infestation. Module grey60, associated with aphid infestation and conventional fertilizer treatment, has Zeaxanthin Epoxidase (Solyc02g090890.3) as its hub gene; this gene is the first step in the abscisic acid biosynthetic pathway [52]. Combined with the enrichment of ABA-related genes in the leaf module blue, this potentially reflects a central role for ABA in response to aphid infestation in this study. Much like module blue in leaf tissue, enrichment of catabolism-related functions is apparent in this module (Fig 11).

In both fertilizer treatment groups, aphid infestation elevated the concentration of phenolic compounds, a compound class with potential implications for human nutrition. Fruit phenolic content appeared to be elevated under the conditions of aphid infestation, yet most of the key, rate-limiting enzymes in the phenolic biosynthesis pathway appear to be suppressed. Accumulation of phenolic compounds is highest earlier in tomato fruit development [53], so it is likely that whatever aphid-host interaction yielded this elevation in phenolic compounds was not present at the time of harvest.

Experimental factors may have contributed to the ability of the studied tomato plants to resist aphid feeding beyond real-world growing conditions. Aphids are primarily limited by the amino acids available in plant phloem [13], which in turn is limited by the nitrogen that can be taken up by the roots and assimilated by the plant. Both treatment groups were regularly fertilized, and the plants given conventional fertilizer in particular had an ample source of readily bioavailable nitrogen. Given more time, the aphids may have been able to overwhelm the ability of the plant to uptake nitrogen and the ability of the plant to transmit resources to its fruit, but in the 9 weeks of infestation provided in this study, this was not the case. Additionally, the extensive damage that aphids can cause in crops is partly the result of pathogens transmitted from plant to plant through their saliva [12]; using pathogen-free aphids limits this destructive aspect of aphid feeding.

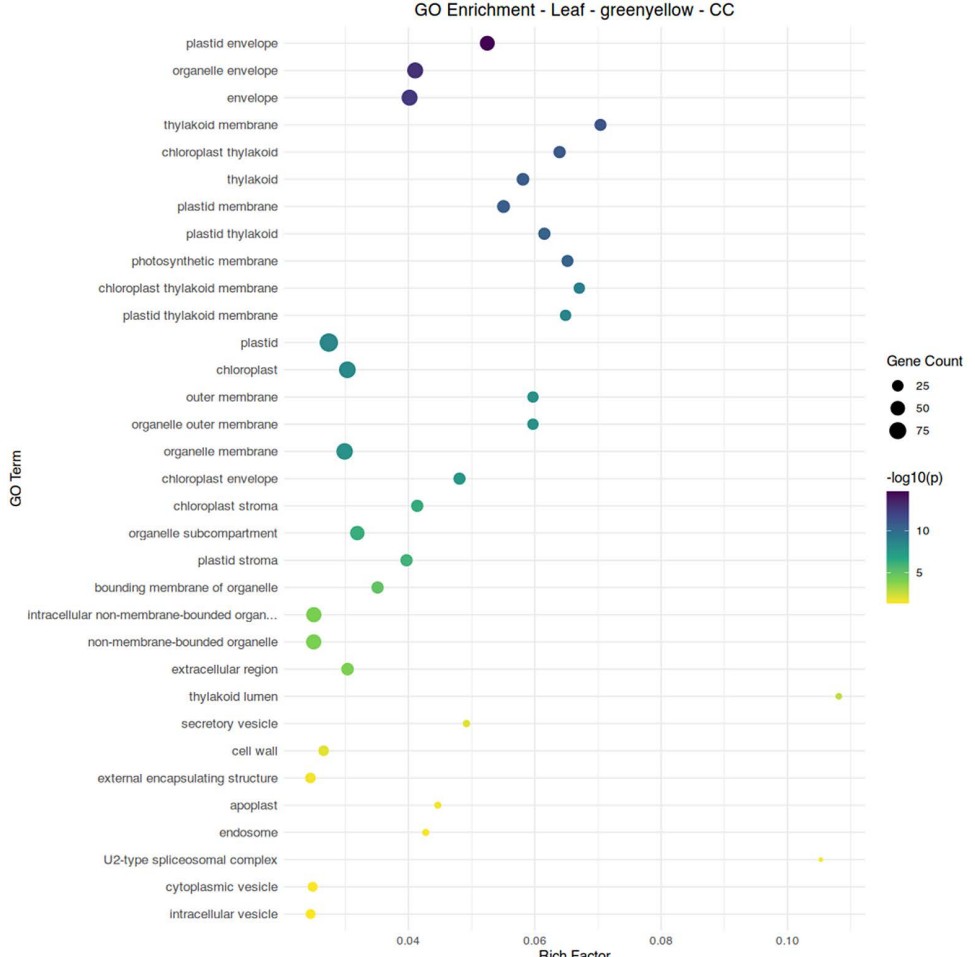

**Fig 5. Gene ontology term enrichment analysis by Fisher's exact test for biological processes in the leaf module turquoise, associated with aphid infestation.** Significance is shown by circle color (scale bar to the right), number of genes are shown by circle size, and rich factor (proportion of genes in the module relative to genes in the annotated dataset) is plotted along the x-axis.

## Conclusion

In this study, tomato plants experiencing uncontrolled aphid infestation were able to maintain their yield and quality through extreme perturbance to their transcriptomic profile and a suppression of endogenous defense responses, regardless of fertilizer type. While infested plants did not experience significant disturbances to the measured biochemical parameters in this experiment, transcriptomic analysis revealed a broad shift from photosynthetic and diverse anabolic processes to catabolism. Module yellow in leaf tissue, and its hub gene (Solyc02g078940.3), may be of interest in exploring the response of plants under organic fertilizer conditions to aphids. Overall plant nitrogen levels may also have been critical in altering the course of infestation. Abscisic acid appears to be the primary hormone response to aphid infestation at the degree of infestation studied, but it does not seem to be associated with resistance to aphids. With the growing demand for organically grown horticultural products, a better understanding of the relationship between fertilization regime, biotic stress resilience, and fruit quality is necessary, especially as organic growing systems are more limited in pest control.

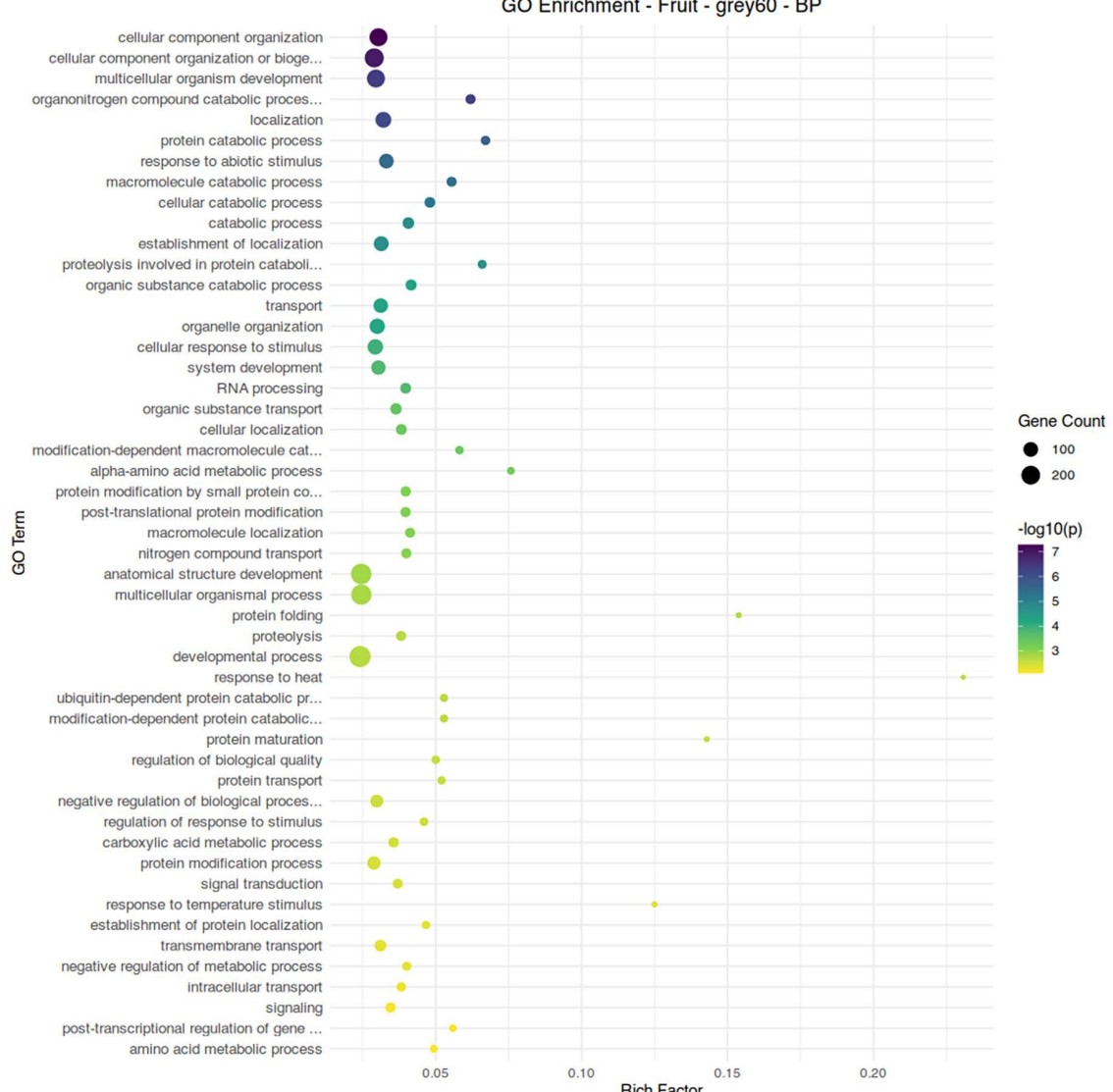

**Fig 6. Gene ontology term enrichment analysis by Fisher's exact test for biological processes in the leaf module blue, associated with the absence of aphid infestation.** Significance is shown by circle color (scale bar to the right), number of genes are shown by circle size, and rich factor (proportion of genes in the module relative to genes in the annotated dataset) is plotted along the x-axis.

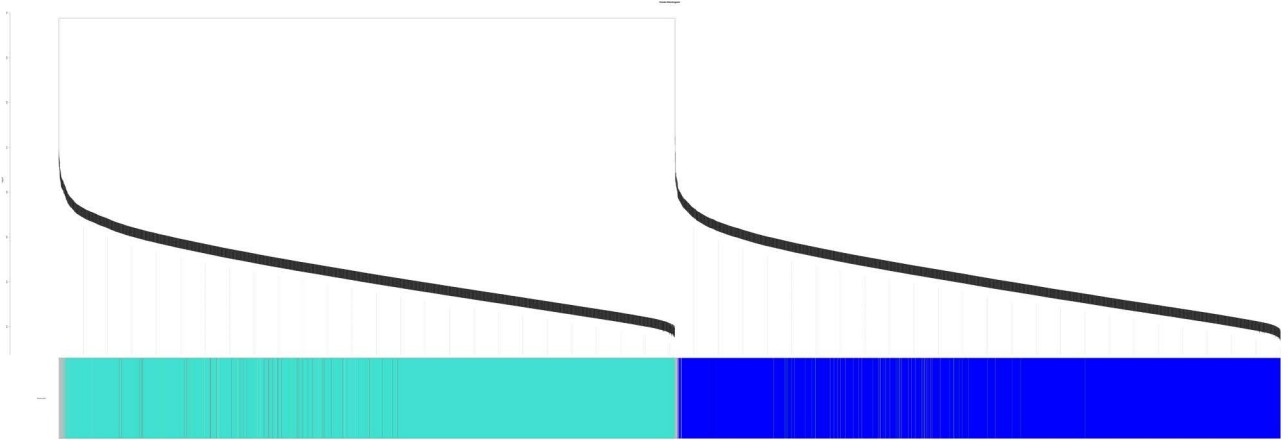

**Fig 7. Gene ontology term enrichment analysis for biological processes in the leaf module yellow, associated with aphid infestation and organic fertilizer treatment.** Significance is shown by circle color (scale bar to the right), number of genes are shown by circle size, and rich factor (proportion of genes in the module relative to genes in the annotated dataset) is plotted along the x-axis.

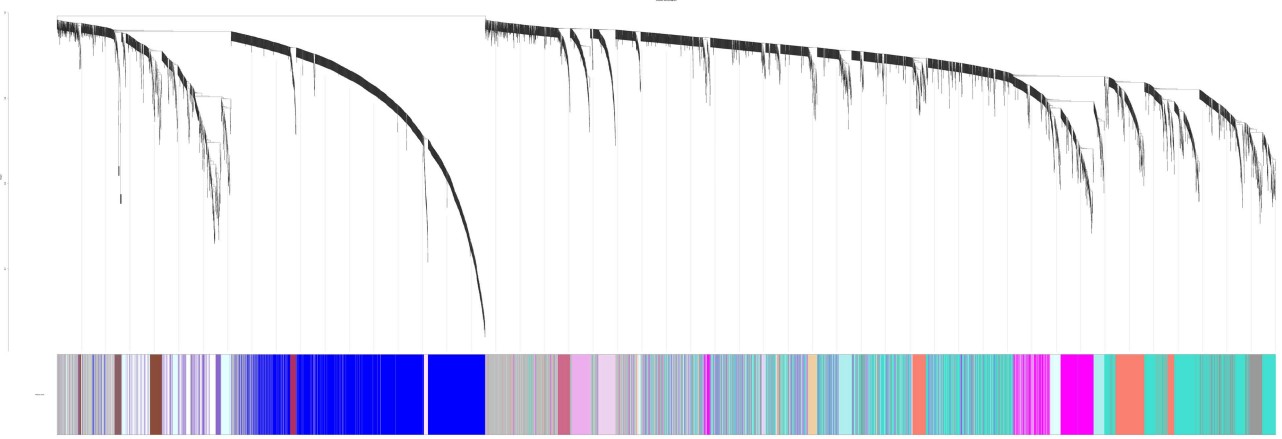

**Fig 8. Gene ontology term enrichment analysis by Fisher's exact test for biological processes in the leaf module red, associated with conventional fertilizer treatment and the absence of aphid infestation.** Significance is shown by circle color (scale bar to the right), number of genes are shown by circle size, and rich factor (proportion of genes in the module relative to genes in the annotated dataset) is plotted along the x-axis.

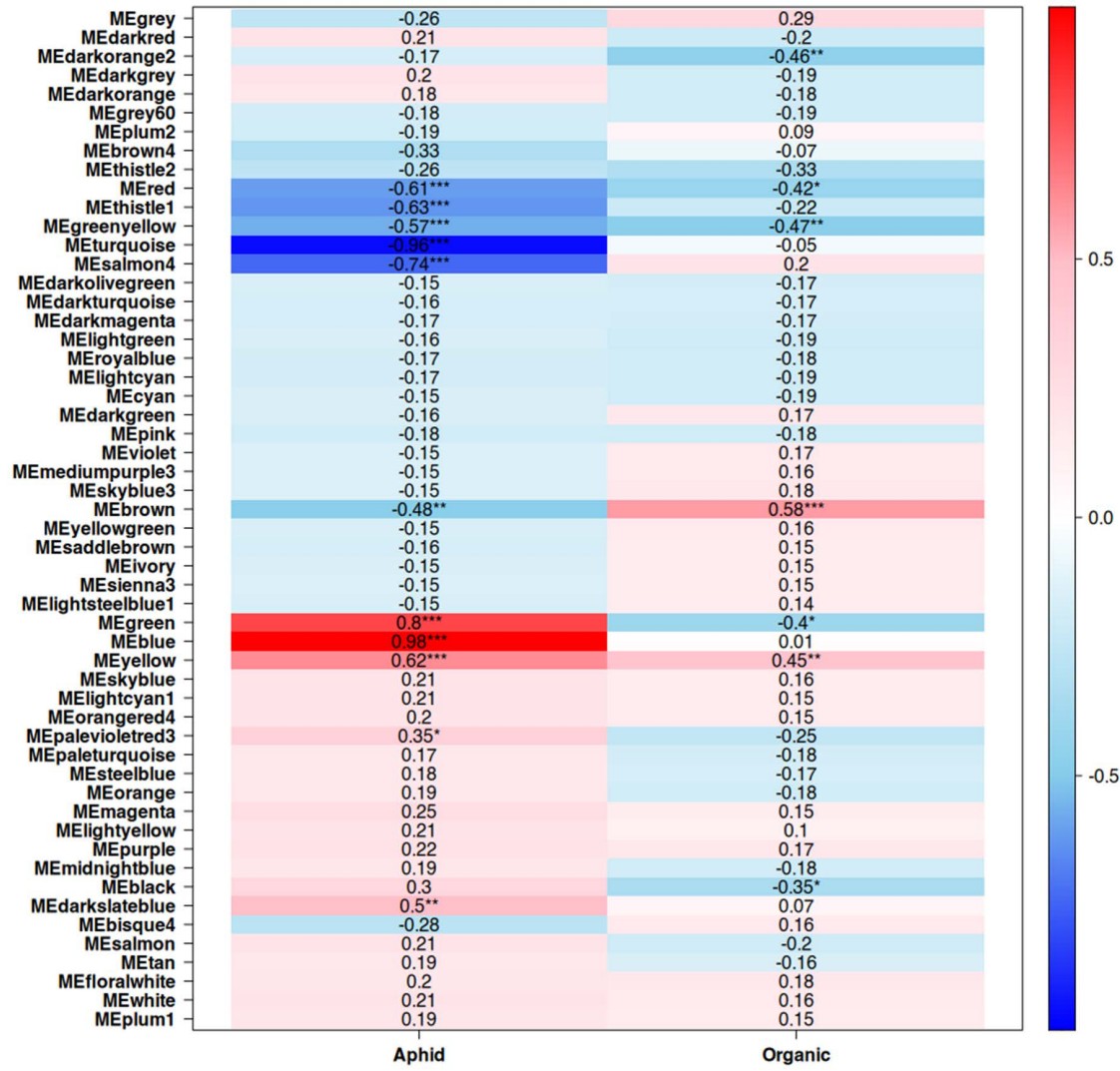

**Fig 9. Gene ontology term enrichment analysis by Fisher's exact test for biological processes in the leaf module greenyellow, associated with conventional fertilizer treatment and the absence of aphid infestation.** Significance is shown by circle color (scale bar to the right), number of genes are shown by circle size, and rich factor (proportion of genes in the module relative to genes in the annotated dataset) is plotted along the x-axis.

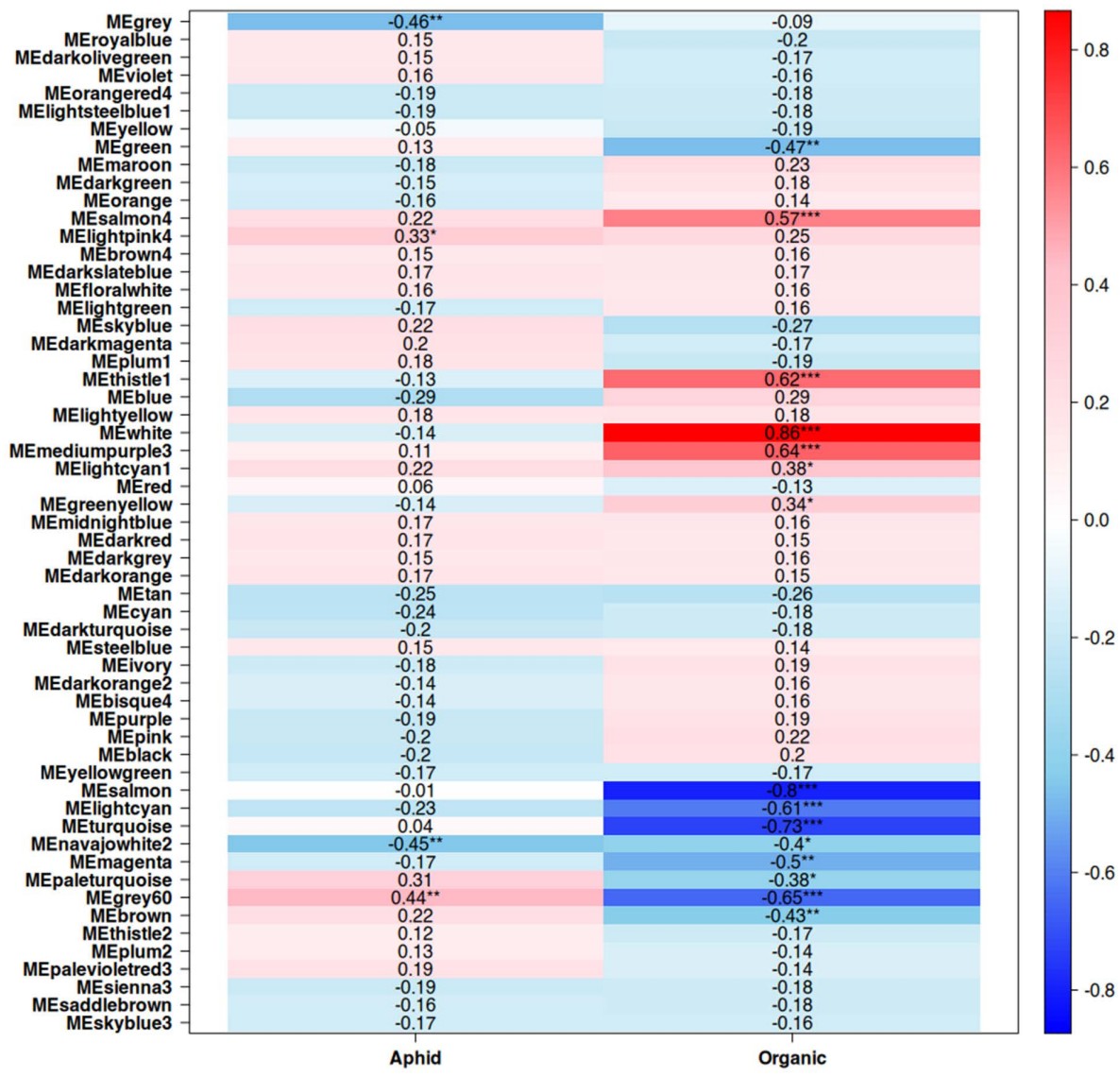

**Fig 10. Gene ontology term enrichment analysis by Fisher's exact test for subcellular localization in the leaf module greenyellow, associated with conventional fertilizer treatment and the absence of aphid infestation.** Significance is shown by circle color (scale bar to the right), number of genes are shown by circle size, and rich factor (proportion of genes in the module relative to genes in the annotated dataset) is plotted along the x-axis.

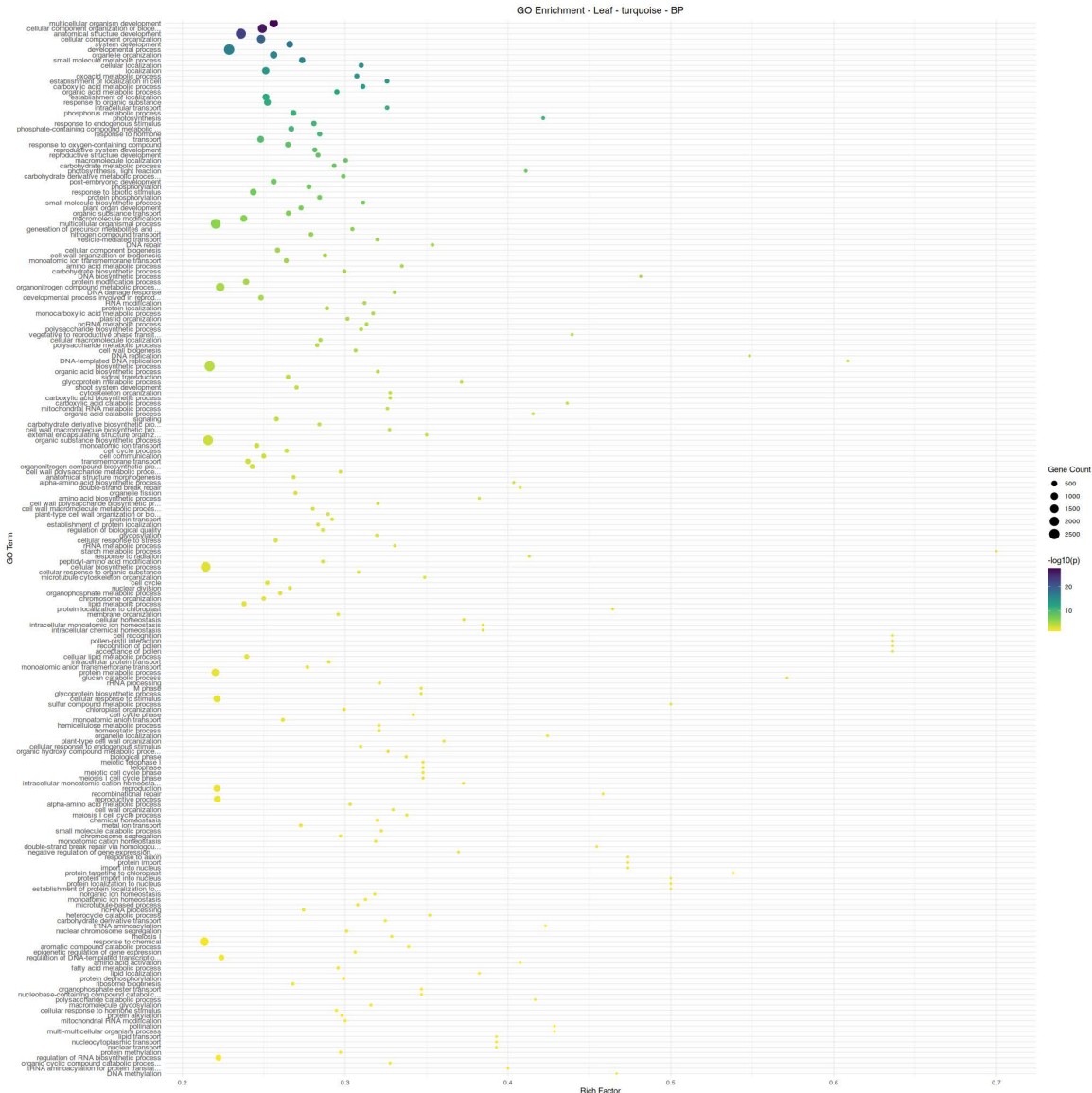

**Fig 11. Gene ontology term enrichment analysis by Fisher's exact test for biological processes in the fruit module grey60, associated with aphid infestation and conventional fertilizer.** Significance is shown by circle color (scale bar to the right), number of genes are shown by circle size, and rich factor (proportion of genes in the module relative to genes in the annotated dataset) is plotted along the x-axis.

## Supporting Information

**S1 Table. Table of gene ontology enrichment data used to generate** Fig 5.
(XLSX)

**S2 Table. Table of gene ontology enrichment data used to generate** Fig 6.
(XLSX)

**S3 Table. Table of gene ontology enrichment data used to generate** Fig 7.
(XLSX)

**S4 Table. Table of gene ontology enrichment data used to generate** Fig 8.
(XLSX)

**S5 Table. Table of gene ontology enrichment data used to generate** Fig 9.
(XLSX)

**S6 Table. Table of gene ontology enrichment data used to generate** Fig 10.
(XLSX)

**S7 Table. Table of gene ontology enrichment data used to generate** Fig 11.
(XLSX)

**S8 Table. Table of exact (to 4 digits) p-values for** Tables 1 **and** 2.
(XLSX)

## Acknowledgments

The authors thank Dr. Richard M. Sharpe, Washington State University for support with data handling.

## Author contributions

**Conceptualization:** Luke Gustafson, Preston Andrews, Amit Dhingra.

**Data curation:** June Labbancz.

**Formal analysis:** June Labbancz, Luke Gustafson.

**Funding acquisition:** Preston Andrews, Amit Dhingra.

**Investigation:** Luke Gustafson, Amit Dhingra.

**Methodology:** June Labbancz, Luke Gustafson, Preston Andrews, Amit Dhingra.

**Project administration:** Preston Andrews, Amit Dhingra.

**Resources:** Preston Andrews.

**Supervision:** Preston Andrews, Amit Dhingra.

**Visualization:** June Labbancz.

**Writing – original draft:** June Labbancz.

**Writing – review & editing:** June Labbancz, Luke Gustafson, Preston Andrews, Amit Dhingra.

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
