## [Decision Letter · Decision Letter 0]

6 Feb 2026

PONE-D-25-66187Both conventionally and organically fertilized tomatoes maintain fruit quality through uncontrolled green peach aphid infestation, with a transcriptional shift towards catabolismPLOS One

Dear Dr. Dhingra,

Thank you for submitting your manuscript to PLOS ONE. After careful consideration, we feel that it has merit but does not fully meet PLOS ONE’s publication criteria as it currently stands. Therefore, we invite you to submit a revised version of the manuscript that addresses the points raised during the review process.

Specifically, your manuscript was reviewed by two independent experts in the field. Both reviewers find the work interesting but raised several issues which need to be addressed properly. The reviewers provide detailed comments in their reviews and pointed out the areas where the manuscript needs to be improved. Therefore, manuscript needs a minor revision to meet the publication standard of PLOS ONE.

We look forward to receiving your revised manuscript.

Kind regards,

Mohammad Irfan, Ph.D.

Academic Editor

PLOS One

“This research was funded in part by CSANR BIOAg grant to PA and AD. USDA National Institute of Food and Agriculture, Hatch projects WNP00011 and Texas A&M AgriLife Hatch Project #TEX0-9950-0 and startup funds from Texas A&M AgriLife Research and Texas A&M University to AD. JL acknowledges graduate research assistantship support from the Department of Horticultural Sciences at Texas A&M University.”

“This research was funded in part by CSANR BIOAg grant to PA and AD. USDA National Institute of Food and Agriculture, Hatch projects WNP00011 and Texas A&M AgriLife Hatch Project #TEX0-9950-0 and startup funds from Texas A&M AgriLife Research and Texas A&M University to AD. JL acknowledges graduate research assistantship support from the Department of Horticultural Sciences at Texas A&M University. The authors thank Dr. Richard M. Sharpe, Washington State University for support with data handling.”

“This research was funded in part by CSANR BIOAg grant to PA and AD. USDA National Institute of Food and Agriculture, Hatch projects WNP00011 and Texas A&M AgriLife Hatch Project #TEX0-9950-0 and startup funds from Texas A&M AgriLife Research and Texas A&M University to AD. JL acknowledges graduate research assistantship support from the Department of Horticultural Sciences at Texas A&M University.”

5. We are unable to open your Supporting Information file Supplementary tables. Please kindly revise as necessary and re-upload.

Reviewers' comments:

Reviewer's Responses to Questions

**Comments to the Author**

1. Is the manuscript technically sound, and do the data support the conclusions?

Reviewer #1: Yes

Reviewer #2: Yes

2. Has the statistical analysis been performed appropriately and rigorously? 

Reviewer #1: Yes

Reviewer #2: No

3. Have the authors made all data underlying the findings in their manuscript fully available?

Reviewer #1: Yes

Reviewer #2: Yes

4. Is the manuscript presented in an intelligible fashion and written in standard English?

Reviewer #1: Yes

Reviewer #2: Yes

5. Review Comments to the Author

Reviewer #1: This manuscript shows Tomatoes produced using both the conventional and organic fertilizer maintained similar yield and quality through uncontrolled green peach aphid infestation, by maintaining biochemical properties and transcriptional shift towards catabolic processes. This study is significant as it extensively assesses the interaction between aphid infestation and organic systems to provide insights on developing improved pest management strategies and identifying targets for crop improvement. The study is well designed and comprehensive as authors compared tomato plants grown in organic or conventional fertilizer with or without aphid infestation and used leaf and fruit tissues for biochemical and transcriptomic profiling. While the manuscript looks technically sound and the data provided supports the conclusion, I have following comments and suggestions to improve the manuscript:

1. Authors stated significantly lower nitrogen content resulting in the lower amino acid availability in the phloem maybe be the limiting factor for aphids in the plants grown in organic fertilizer and supported it with the literature (Mattson WJ 1980 Annu Rev Ecol Systm). I would suggest authors to add more reports to the discussion section on the role of nitrogen and amino acids in aphid infestation of plants.

2. In addition to nutrient and plant defense responses, I would recommend authors to discuss the role of soil microbiome in aphid-plant interactions in organic vs conventional farming practices and provide relevant references.

3. Authors tested the effect of aphid infestation on the yield of tomato plants in conventional vs organic fertilizer (Table 1). I would suggest author to add more information to the material and method sections Line 117 on growth stage of aphids added/counted, timepoint and method of counting.

4. Authors should add more information to the table and figure legends like number of plants (N) and type of statistical method used.

5. The quality of images is very poor in Figure 1 and Figure 2 of the manuscript PDF file and it’s hard to understand the axis labels. This may have been caused during file conversion, as the attached PNG images look good. I would suggest authors to check this and see if they can improve the image quality.

6. Line 232: I would suggest authors to edit the sentence to make it clear that Cluster dendogram for leaf and fruit are shown in Figure1 and Figure 2 respectively.

7. Line 234: Same as line 232, authors should make it clear that correlation plots in the gene co-expression network for leaf and fruit samples are shown in Figure 3 and Figure 4 respectively.

Reviewer #2: 1. The manuscript would benefit from clearer reporting of the statistical framework used for trait comparisons. Please specify the exact model structure (e.g., two-way ANOVA with fertilizer and aphid treatment as fixed effects), confirm whether interaction terms were tested throughout, and indicate how assumptions (normality, homoscedasticity) were evaluated. Where feasible, reporting exact p-values and degrees of freedom (possibly in supplementary material) would improve transparency.

2. Several figure captions, particularly for module–trait correlation heatmaps and GO enrichment analyses, lack sufficient detail to be interpreted independently. Please clarify color scales, significance thresholds, and whether GO enrichment results are corrected for multiple testing. Additionally, terminology such as “organic fertilizer treatment,” “organic production,” and “organic growing system” is used interchangeably throughout the manuscript; standardizing this language would improve clarity.

6. PLOS authors have the option to publish the peer review history of their article (what does this mean?). If published, this will include your full peer review and any attached files.

Reviewer #1: No

Reviewer #2: **Yes:** RAMGOPAL PRAJAPATI

---

## [Author Response · Author response to Decision Letter 1]

13 Apr 2026

April 13, 2026

Dear Editor,

We have addressed the feedback received from the reviewers and the recent edits requested by the editorial office. A point-by-point response follows.

Best regards,

Amit Dhingra, Ph.D.

Corresponding author

April 12, 2026

Edits requested by Editorial Office

1.Thank you for stating the following financial disclosure:

“This research was funded in part by CSANR BIOAg grant to PA and AD. USDA National Institute of Food and Agriculture, Hatch projects WNP00011 and Texas A&M AgriLife Hatch Project #TEX0-9950-0 and startup funds from Texas A&M AgriLife Research and Texas A&M University to AD. JL acknowledges graduate research assistantship support from the Department of Horticultural Sciences at Texas A&M University.”

Response: As advised the cover letter has been updated with the following text: The funders had no role in study design, data collection and analysis, decision to publish, or preparation of the manuscript.

2.We note that you have provided funding information that is currently declared in your Funding Statement. However, funding information should not appear in the Acknowledgments section or other areas of your manuscript. We will only publish funding information present in the Funding Statement section of the online submission form.

Response: The manuscript has been edited and funding information has been removed from the Acknowledgements section.

3.We are unable to open your Supporting Information file TableS1.tsv, TableS2.tsv, TableS3.tsv, TableS4.tsv, TableS5.tsv, TableS6.tsv, TableS7.tsv, TableS8.csv. Please kindly revise as necessary and re-upload.

Response: All the .tsv files can be opened using Excel. However, for convenience, the files have been converted to .xlsx format. The .xlsx format files have been uploaded as requested.

Revision 1

Point-by-point response to reviewer’s comments:

Please note: Line numbers correspond to the Track Changes version of the manuscript.

Reviewer #1: This manuscript shows Tomatoes produced using both the conventional and organic fertilizer maintained similar yield and quality through uncontrolled green peach aphid infestation, by maintaining biochemical properties and transcriptional shift towards catabolic processes. This study is significant as it extensively assesses the interaction between aphid infestation and organic systems to provide insights on developing improved pest management strategies and identifying targets for crop improvement. The study is well designed and comprehensive as authors compared tomato plants grown in organic or conventional fertilizer with or without aphid infestation and used leaf and fruit tissues for biochemical and transcriptomic profiling. While the manuscript looks technically sound and the data provided supports the conclusion, I have following comments and suggestions to improve the manuscript:

1. Authors stated significantly lower nitrogen content resulting in the lower amino acid availability in the phloem maybe be the limiting factor for aphids in the plants grown in organic fertilizer and supported it with the literature (Mattson WJ 1980 Annu Rev Ecol Systm). I would suggest authors to add more reports to the discussion section on the role of nitrogen and amino acids in aphid infestation of plants.

Response: As suggested, discussion of the relationship between plant nitrogen status and aphid feeding success has been expanded upon and the corresponding paragraph has been modified. Additional references have also been added. See lines 357-368.

2. In addition to nutrient and plant defense responses, I would recommend authors to discuss the role of soil microbiome in aphid-plant interactions in organic vs conventional farming practices and provide relevant references.

Response: The authors agree that soil microbiome is an important aspect in plant response to stressors and works cooperatively with fertilization regime in modulating plant health. Mention of this has been added in the introduction, see lines 99-100. As no experimental work on soil microbiome was conducted in this study, extensive discussion would be beyond the scope of this work.

3. Authors tested the effect of aphid infestation on the yield of tomato plants in conventional vs organic fertilizer (Table 1). I would suggest author to add more information to the material and method sections Line 117 on growth stage of aphids added/counted, timepoint and method of counting.

Response: As per reviewer feedback, additional information on the growth stage of the aphids, their quantification, and the time of their quantification was added to the manuscript. See lines 124-126 and 128-131.

4. Authors should add more information to the table and figure legends like number of plants (N) and type of statistical method used.

Response: We have added more details to the table and figure legends regarding sample sizes and statistical analyses conducted. See lines 204-209; 221; 244-246; 249-253; 270-271; 273-274; 294-303; 320-323; 334-348; 380-384.

5. The quality of images is very poor in Figure 1 and Figure 2 of the manuscript PDF file and it’s hard to understand the axis labels. This may have been caused during file conversion, as the attached PNG images look good. I would suggest authors to check this and see if they can improve the image quality.

Response: The reviewer is correct. High resolution images have been submitted to the journal.

6. Line 232: I would suggest authors to edit the sentence to make it clear that Cluster dendogram for leaf and fruit are shown in Figure1 and Figure 2 respectively.

Response: Edited to clarify this in the text, in addition to the existing captions for Figures 1 and 2. See line 258.

7. Line 234: Same as line 232, authors should make it clear that correlation plots in the gene co-expression network for leaf and fruit samples are shown in Figure 3 and Figure 4 respectively.

Response: Edited this sentence as well, in addition to the existing captions for Figures 3 and 4. See lines 260.

Reviewer #2

1. The manuscript would benefit from clearer reporting of the statistical framework used for trait comparisons. Please specify the exact model structure (e.g., two-way ANOVA with fertilizer and aphid treatment as fixed effects), confirm whether interaction terms were tested throughout, and indicate how assumptions (normality, homoscedasticity) were evaluated. Where feasible, reporting exact p-values and degrees of freedom (possibly in supplementary material) would improve transparency.

Response: Please see the response to the other reviewer. Briefly, information on the statistical frameworks used for our analysis has been elaborated upon, as well as the specific software used to process the data and evaluate and transform the data in response to assumptions. Wherever log transformation was necessary, it has been clarified in the text. See lines 206 and 252. Overall additional details have been included in table and figure captions, as well. See lines 204-209; 221; 244-246; 249-253; 270-271; 273-274; 294-303; 320-323; 334-348; 380-384. Interaction terms were tested throughout, the results of which can be seen in Tables 1 and 2. Exact p-values to 4 digits for Tables 1 and 2 have been provided as a new supplementary material - Table S8.

2. Several figure captions, particularly for module–trait correlation heatmaps and GO enrichment analyses, lack sufficient detail to be interpreted independently. Please clarify color scales, significance thresholds, and whether GO enrichment results are corrected for multiple testing. Additionally, terminology such as “organic fertilizer treatment,” “organic production,” and “organic growing system” is used interchangeably throughout the manuscript; standardizing this language would improve clarity.

Response: Additional details have been added to figure captions to assist independent interpretation, See lines 204-209; 221; 244-246; 249-253; 270-271; 273-274; 294-303; 320-323; 334-348; 380-384.

Multiple testing was not conducted for GO enrichment results as the GO hierarchy violates the assumption of independence of tests. The statistical test used for this work has been clarified, see lines 219-221.

Differentiation between organic fertilization and organic fertilizer has been made throughout the manuscript to clarify the fact that this study only evaluates an organic input, rather than other aspects of organic production and fertilization. In the introduction, however, the term “organic production” is used more broadly since that section discusses organic production in a general context.

---

## [Editor Report · Decision Letter 1]

21 Apr 2026

Both conventionally and organically fertilized tomatoes maintain fruit quality through uncontrolled green peach aphid infestation, with a transcriptional shift towards catabolism

PONE-D-25-66187R1

Dear Dr. Dhingra,

We’re pleased to inform you that your manuscript has been judged scientifically suitable for publication and will be formally accepted for publication once it meets all outstanding technical requirements.

Kind regards,

Mohammad Irfan, Ph.D.

Academic Editor

PLOS One
---

## [Editor Report · Acceptance letter]

PONE-D-25-66187R1

PLOS One

Dear Dr. Dhingra,

I'm pleased to inform you that your manuscript has been deemed suitable for publication in PLOS One. Congratulations! Your manuscript is now being handed over to our production team.

Kind regards,

on behalf of

Dr. Mohammad Irfan

Academic Editor

PLOS One